# Multi-Objective Optimization of Dimensional Accuracy in Electric Hot Incremental Sheet Forming

**Zhengfang Li [1], Songlin He [1], Zhiguo An [2], Zhengyuan Gao [2,*] and Shihong Lu [3]**

1   School of Mechanical and Electrical Engineering, Kunming University, Kunming 650214, China;
    willienuaa@163.com (Z.L.)
2   School of Mechanotronics and Vehicle Engineering, Chongqing Jiaotong University, Chongqing 400074, China;
    azgcqu@163.com
3   College of Mechanical & Electrical Engineering, Nanjing University of Aeronautics and Astronautics,
    Nanjing 210016, China
*   Correspondence: zhengyuangao@cqjtu.edu.cn

**Abstract:** The forming defects of parts mainly include fracture, springback, thermal expansion, and rough surface during electric hot incremental forming, in which the springback and thermal expansion directly affects the dimensional accuracy of parts. In this paper, the combined optimization of process parameters and unsupported distances was proposed to control the dimensional accuracy of forming parts in the electric hot incremental sheet forming process. The predictive model of major errors was established through the response surface methodology, and then the multi-objective optimal model was obtained using Non-dominated Sorting Genetic Algorithms-II (NSGA-II). Meanwhile, a multi-objective optimal result was determined according to the error compensation feature of the forming process. On this basis, the effect of unsupported distances on dimensional errors was analyzed in detail, and a valid unsupported distance was proposed to further improve the forming accuracy of the whole forming region. Finally, the experimental result demonstrated that the combined optimal method proposed was accurate and feasible.

**Keywords:** incremental sheet forming; electric hot forming; multi-objective optimization; dimensional accuracy control; error prediction





## 1. Introduction

The layer machining characteristic of incremental sheet forming can improve the formability of materials. However, the springback of forming parts is largely due to this machining characteristic, which leads to a lower dimensional accuracy in the whole process, and then the development of incremental forming is restricted. In order to overcome the above defect, a series of studies such as structure design [1–3], accuracy control [4,5], parameters optimization [6–8], forming strategy design [9,10] and others are implemented to enhance the forming quality of parts. Allwood et al. [1,2] points out that the dimensional error of incremental forming is about ±3 mm and it is divided into three portions: the clamping error, the non-clamping error, and the final error. In order to improve the key problem of low dimensional accuracy of parts, two main methods, namely, adding auxiliary support and tool path compensation, are proposed to enhance the manufacturing accuracy of parts [4]. Although various assistant methods [5] are designed in the incremental sheet forming process, the use of these methods can promote the complexity of the whole process and the fabricating cost of parts. Therefore, path optimization is still a main way to improve the dimensional accuracy of forming parts.

With the development of industry, the application of light alloys is gradually increasing. However, the alloy has a poor plasticity at room temperature and has a good plastic property at high temperature. Therefore, traditional incremental sheet forming cannot be adopted to fabricate the part with light alloys. According to the above issue, hot incremental

sheet forming processes (shown in Figure 1) such as laser incremental sheet forming [11], thermal medium incremental sheet forming [12], electric hot increment sheet forming (EHIF) [13,14], and others, are proposed in which electric hot incremental sheet forming is widely used to fabricate light alloy parts due to the fact that the forming process has a high heating efficiency and a simple device structure. Aiming at the electric hot incremental forming (EHIF) of light alloys, the forming quality of parts mainly includes the surface roughness and the dimensional accuracy., in which the improved method of the surface roughness mainly includes process optimization [15,16] and self-lubricant technology [17]. In addition to this, the effect factor of dimensional accuracy is more complex than that of conventional incremental sheet forming, especially the interaction between process parameters and thermal expansion [18–20]. Fan et al. [21] adopted a composite process of reverse drawing and electric hot incremental forming to limit the reverse bulge at the bottom of titanium alloy parts, and Ambrogio et al. [22] further discussed the relationship between forming limit angle and current density for EHIF of AA2024-T3, AZ31B-O, and Ti-6Al-4V alloys, and the corresponding limit curve was established, which could provide a reliable reference for EHIF of hard-to-form sheet metals. Skjoedt [23] and Shi [24] separately proposed the algorithm of machining paths to improve the rate of product finished and the forming quality. Currently, the process parameter optimization is usually adopted to improve the forming accuracy of parts [25].

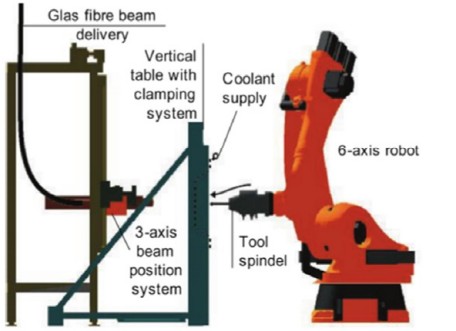
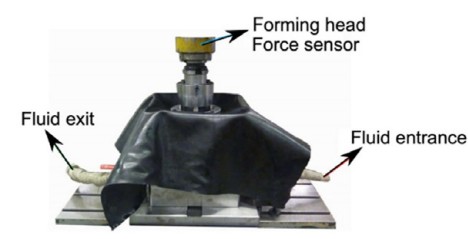

**Laser incremental sheet forming**  **Thermal medium incremental sheet forming**

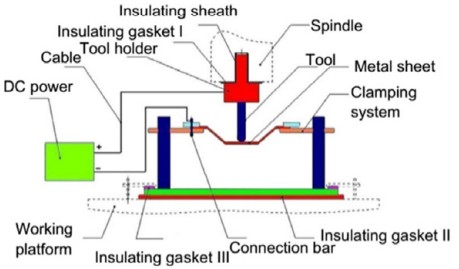
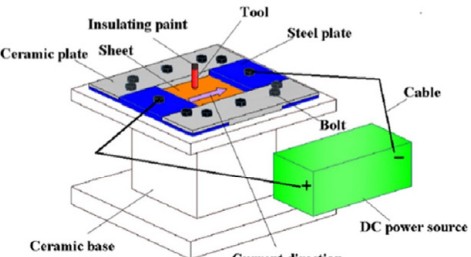

**Electric hot increment sheet forming**

**Figure 1.** The typical hot incremental sheet forming process.

According to the above study, the improvement method of dimensional accuracy is mainly the optimization of three aspects: process parameters, forming process, and machining path. However, the interaction between process parameters and unsupported distance is usually not considered in the improvement of dimensional accuracy. In this paper, the combined optimization of process parameters and unsupported distances was proposed to control the dimensional accuracy of forming parts in EHIF. The predictive model of the two major targets was separately established through the response surface methodology, and the corresponding multi-objective optimal model was obtained by Non-dominated Sorting Genetic Algorithms-II (NSGA-II). On this basis, the effect of unsupported distances

on dimensional errors was further analyzed, and then a reasonable forming scheme was proposed to obtain a part within ±0.5 mm error. Meanwhile, an engineering part was successfully fabricated based on the optimal forming scheme, which further verified the feasibility of the aforementioned optimal method.

## 2. Materials and Methods

According to the above analysis, the effect of unsupported region on dimensional accuracy is easy to be ignored in EHIF, which is shown in Figure 2. In this work, a square cone with Ti-6Al-4V titanium alloy was designed to analyze the effect of process parameters and unsupported distances on dimensional accuracy, the dimensions of which were: 70 mm opening size, 45° forming angle, and 15 mm forming height. The sheet thickness was 0.5 mm and the whole forming process was carried out in an LNC-M700 machine, and the details of the setup are carefully described in Figure 3. The direct current power with 0–1500 A was adopted to provide the heat for the deformation region, and a thermal imager (the collecting range of −20 to 1500 °C and the collecting error of ±0.1 °C) was used to ensure the forming temperature. Meanwhile, a spiral machining path was designed to fabricate the part and to eliminate the machining trace of the surface.

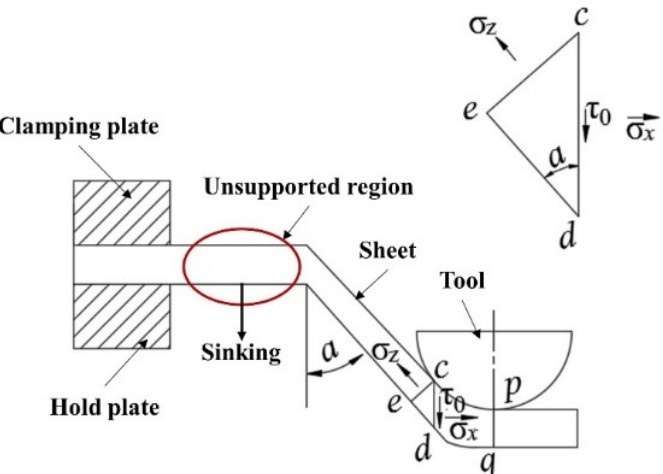

**Figure 2.** Sketch of unsupported and forming regions.

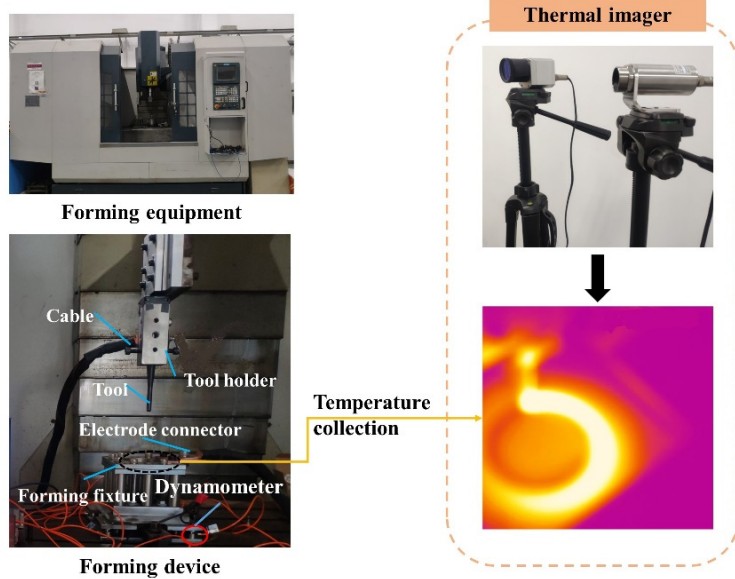

**Figure 3.** The test setup of EHIF.

According to the simulation results (Figure 4), the dimensional error of forming parts was mainly distributed in two portions; upper and lower fillets. Therefore, the two portions are viewed as optimal goals, expressed in *A1* and *A2* respectively. According to the study of Honarpisheh [26], the tool diameter (*d*), the step size (*z*), the feed rate (*f*), and the current gradient (*g*) were major influence factors for the dimensional error of parts in EHIF, and the current gradient was a significant factor in ensuring the temperature of the deformation region when the forming temperature was set at 500 °C. Meanwhile, the interaction between the tool diameter and the step size was greater than the effect of a single factor; additionally, the combination (*d/z*) of tool diameter and step size was viewed as a factor. According to the response surface design analysis method [27], three factors with three levels were designed through the normalization method and shown in Table 1.

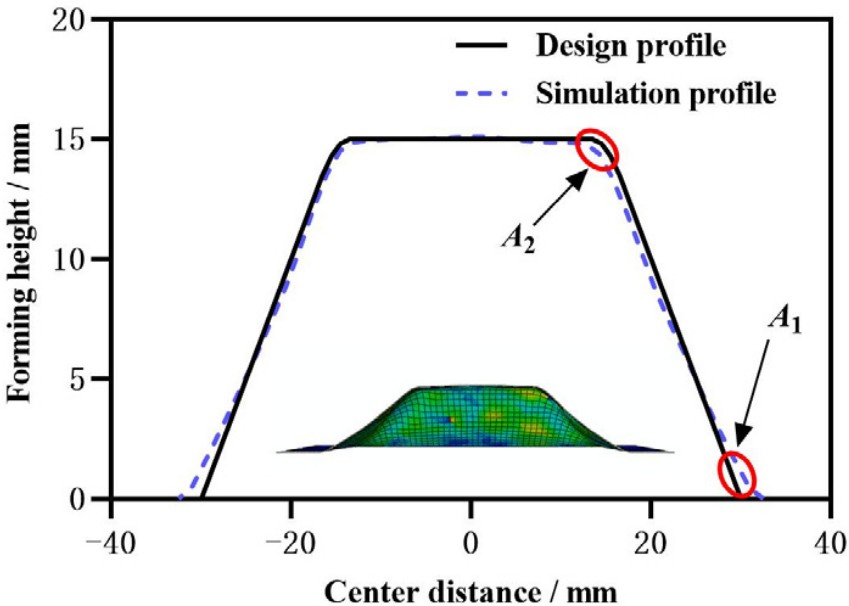

**Figure 4.** Distribution of the main dimension errors of parts.

**Table 1.** Factors and levels of experimental design.

| Code | Factor | Level | | |
|---|---|---|---|---|
| | | −1 | 0 | 1 |
| *a* | *d/z* (mm) | 8/0.1 | 10/0.15 | 12/0.2 |
| *b* | *f* (mm·min$^{-1}$) | 500 | 650 | 800 |
| *c* | *g* (A) | 30 | 40 | 50 |

According to the above analysis, 17 test groups (Table 2) were designed to analyze the effect of process parameters on the dimensional accuracy based on the response surface methodology, and the corresponding error prediction model was established to obtain populations of the multi-objective optimization. On this basis, five unsupported distances, 0.5 mm, 1 mm, 1.5 mm, 2 mm, and 2.5 mm, were designed to analyze the change rule of *A1*. Then, the optimal combination of process parameters and unsupported distances was obtained to fabricate the part within ± 0.5 mm error. The corresponding procedure of the multi-objective optimization is shown in Figure 5. According to the above analysis, the optimal model was further established based on the NSGA-II and it is written as:

$$\begin{cases} \min A1(\mathrm{x}) = G_1 \\ \min A2(\mathrm{x}) = G_2 \end{cases} \tag{1}$$

where $G1$ and $G2$ are (separately) the optimal solution set of $A1$ and $A2$. The corresponding constraint condition was the range of $-1$ to 1 for each factor. Firstly, the random function was adopted to obtain the initial population based on the predictive models of $A1$ and $A2$, and the offspring population of the initial sample was further obtained using the non-dominated sorting method. Secondly, the individual severity of crowding was screened to obtain the new parent using the non-dominated sorting method. By analogy, the selection-crossover-mutation process was used to obtain the offspring data sequence until the constraint conditions of the program, namely, the optimal solution set of targets, were met. NSGA-II analyzed nonlinear and discontinuous multi-objective optimization problems, and it utilized the small mirror allocation method to optimize the entire designed space, and then the optimal solution set had a feature of uniform distribution. Therefore, the adaptability and the diversity of the population were further improved. In addition to this, the selection algorithm included the hierarchical fast non-dominated sorting method and peer crowding screening technology, which greatly improved the computational efficiency of the optimal model. Meanwhile, the elite individuals, namely, non-dominated individuals, were retained using the screening method, and the individuals were added to the next generation population. In this way, the genetic individual superiority was ensured.

**Table 2.** The experimental design scheme and the results of EHIF.

| No. | $a$ | $b$ | $c$ | $A1$/mm | $A2$/mm |
|-----|------|------|------|---------|---------|
| 1 | 1.00 | −1.00 | 0.00 | 1.24 | 0.28 |
| 2 | −1.00 | 1.00 | 0.00 | 1.67 | 0.14 |
| 3 | 1.00 | 1.00 | 0.00 | 1.25 | 0.34 |
| 4 | 0.00 | 0.00 | 0.00 | 1.26 | 0.31 |
| 5 | 0.00 | −1.00 | −1.00 | 1.25 | 0.18 |
| 6 | 0.00 | 0.00 | 0.00 | 1.26 | 0.31 |
| 7 | 0.00 | 0.00 | 0.00 | 1.26 | 0.31 |
| 8 | −1.00 | −1.00 | 0.00 | 1.25 | 0.13 |
| 9 | −1.00 | 0.00 | 1.00 | 1.55 | 0.27 |
| 10 | 0.00 | 0.00 | 0.00 | 1.26 | 0.31 |
| 11 | 0.00 | 0.00 | 0.00 | 1.26 | 0.31 |
| 12 | 0.00 | 1.00 | 1.00 | 1.39 | 0.44 |
| 13 | 1.00 | 0.00 | 1.00 | 1.36 | 0.14 |
| 14 | 0.00 | 1.00 | −1.00 | 1.39 | 0.44 |
| 15 | −1.00 | 0.00 | −1.00 | 1.55 | 0.27 |
| 16 | 0.00 | −1.00 | 1.00 | 1.25 | 0.18 |
| 17 | 1.00 | 0.00 | −1.00 | 1.36 | 0.14 |

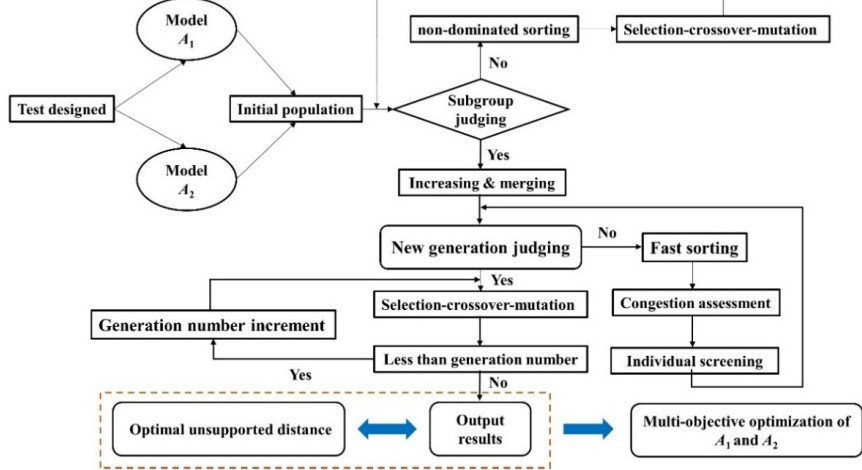

**Figure 5.** The multi-objective optimization procedure of EHIF.

## 3. Results and Discussion

### 3.1. Establishment of the Prediction Model

The second-order model was adopted to separately established the initial regressive model of $A1$ and $A2$ according to the experimental results of Table 2, and the variance analysis method was used to estimate the significance of each term for the regressive model. The $p$-value of 0.05 was viewed as an evaluable standard, namely that the $p$-value of each term needed to be less than 0.05. Based on the aforementioned method, the experimental values of $A1$ and $A2$ were separately collected according to the experimental data of Table 2, and the corresponding initial predictive model was established through the regression fitting method. On this basis, the $p$-value of each term of initial models was analyzed, and the nonsignificant term, namely that the $p$-value of the term was greater than 0.05, could be eliminated. The modified model was then established and evaluated based on the reserved terms. According to the above process, the initial models were continually simplified until the main terms of models were significant. The final predictive models were separately obtained through the aforementioned method and are written as:

$$A_1 = 1.25 - 0.1a + 0.089b - 0.1ab + 0.11a^2 + 0.08c^2 \tag{2}$$

$$A_2 = 0.34 + 0.03a + 0.0075b - 0.17a^2 + 0.092b^2 \tag{3}$$

Table 3 shows significant terms of $A1$ and $A2$, in which the interaction between a and b was significant for $A1$, as shown in Figure 6. The interaction between $a$ and $b$ was similar to a saddle-shaped surface, and the changing trend of the target value was first increased and then decreased along the diagonal composed of point $(-1, -1)$ and point $(1, 1)$, and the changing trend was contrary along the diagonal composed of point $(-1, 1)$ and point $(1, 1)$. Meanwhile, the maximum and minimum values were obtained at points $(1, -1)$ and $(-1, -1)$, respectively. In general, the value of $A1$ fluctuated with the change of a and b, which showed that the two process parameters had an obvious interaction with $A1$. The model of $A2$ had two significant terms, $a^2$ and $b^2$. Although the terms $a$ and $b$ were not significant, the two terms had to be reserved due to the fact that the second-order term relied on the corresponding single term. Therefore, the model of $A2$ needed to include $a$ and $b$. In addition to this, the term c of the model $A2$ could be ignored, as the effect of c on $A2$ was non-significant. The region of $A2$ was the bottom of forming parts and its forming temperature was affected by thermal superposition. The forming temperature of $A2$ was higher than that of the initial forming region. Therefore, the effect of c on $A2$ was non-significant and ignored.

**Table 3.** Significant terms of $A1$ and $A2$.

| $A_1$ | Sum Sq. | Df | Mean Sq. | F-Value | $p$-Value | Significance |
|---|---|---|---|---|---|---|
| $a$ | 0.082 | 1 | 0.082 | 163.17 | 0.0001 | ** |
| $b$ | 0.063 | 1 | 0.063 | 125.37 | 0.0001 | ** |
| $ab$ | 0.042 | 1 | 0.042 | 83.61 | 0.0001 | ** |
| $a^2$ | 0.054 | 1 | 0.054 | 106.56 | 0.0001 | ** |
| $c^2$ | 0.027 | 1 | 0.027 | 53.94 | 0.0001 | ** |
| $A_2$ | Sum Sq. | Df | Mean Sq. | F-value | $p$-value | Significance |
| $a^2$ | 0.13 | 1 | 0.13 | 22.23 | 0.0005 | ** |
| $b^2$ | 0.035 | 1 | 0.035 | 6.20 | 0.0284 | * |

Note: ** ($p < 0.01$) and * ($p < 0.05$).

Figures 7 and 8 separately show the adaptability of the predictive models of $A1$ and $A2$, in which the standardized residuals of each model both conformed to normal distribution, and the other three indexes also fluctuated within the allowable range. Therefore, the predictive models of $A1$ and $A2$ were accurate and reliable according to the above analysis.

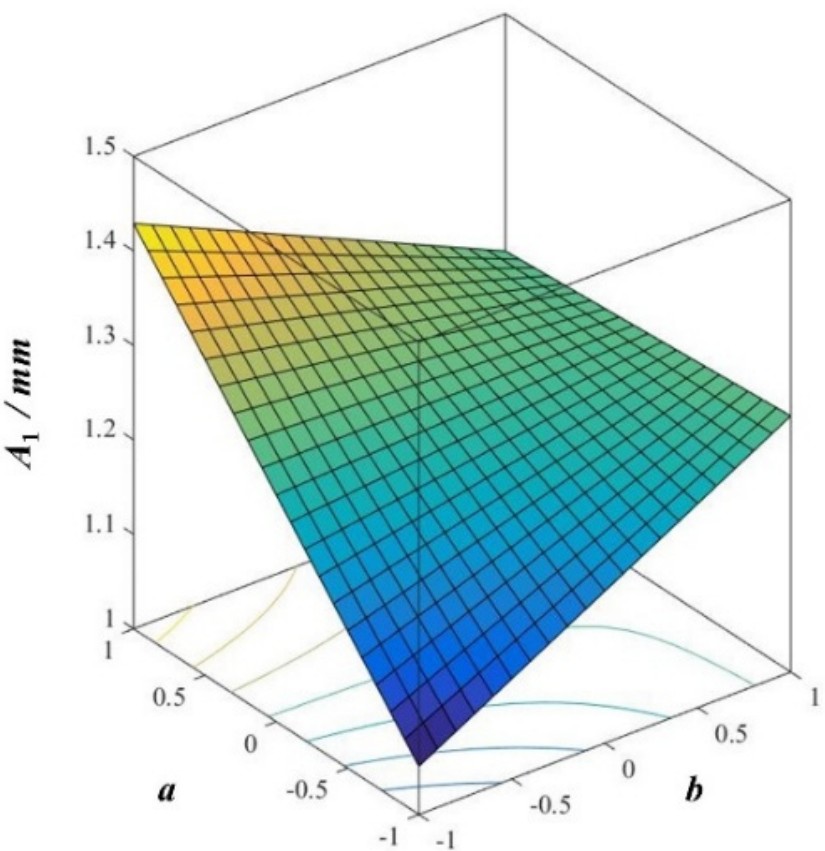

**Figure 6.** The response surface of the interaction between a and b.

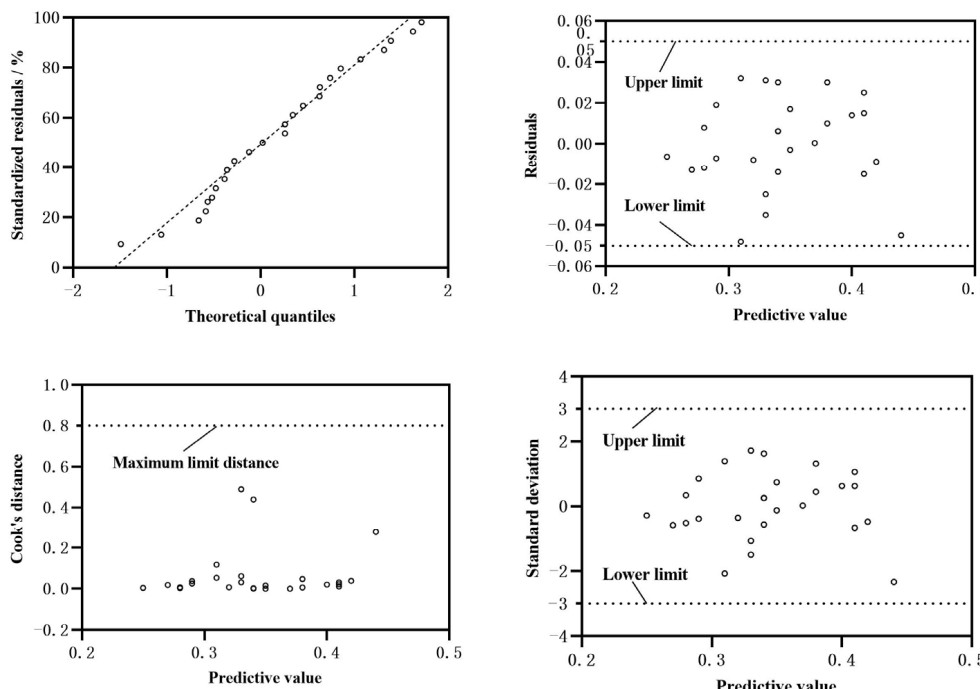

**Figure 7.** The adaptability analysis of the predictive model of *A*1.

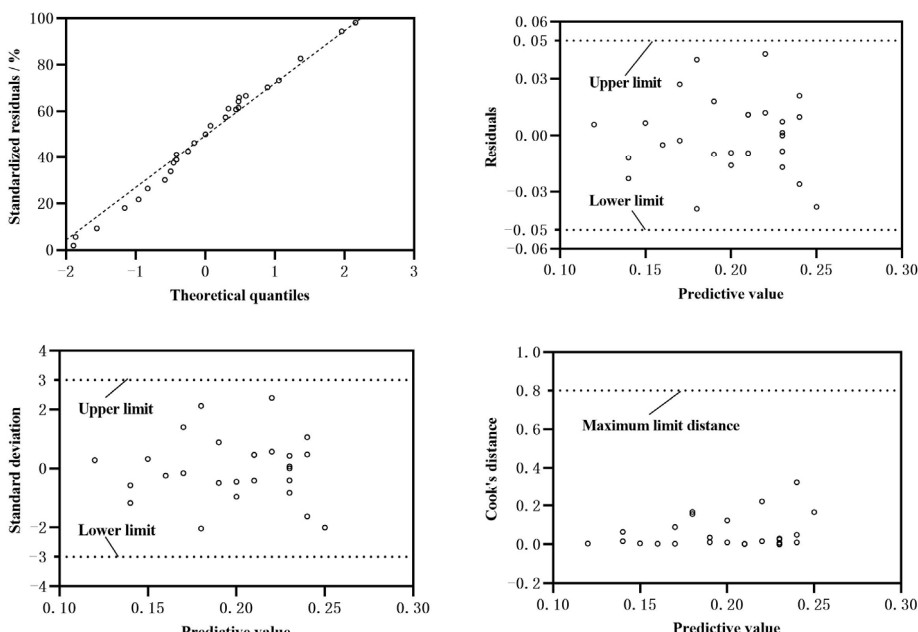

**Figure 8.** The adaptability analysis of the predictive model of *A*2.

### 3.2. Effect of Optimal Parameters

Figure 9 shows the effect of the initial population on the optimal accuracy, in which the optimized results of (*A*1, *A*2) were (1.45, 0.14) and (1.16, 0.42) when the initial population size was 50. According to the above results, the singularity between *A*1 and *A*2 was existent: *A*1 was maximum when *A*2 was minimum. The minimum of *A*2 needed to ensure a minimum since the compensation of *A*2 was difficult. Therefore, the minimum point of *A*2 was viewed as the optimal solution set. The results of four groups (1.45, 0.14) and (1.16, 0.42), (1.44, 0.14) and (1.17, 0.42), (1.44, 0.14) and (1.16, 0.42), and (1.45, 0.14) and (1.16, 0.42) were obtained when the initial population size was separately set as 100, 150, 200, and 250. According to the above analysis, the errors between optimal results were both less than 5%, and the effect of initial populations on optimal accuracy was not significant. Therefore, the initial population size of 50 was selected in order to reduce the optimal time.

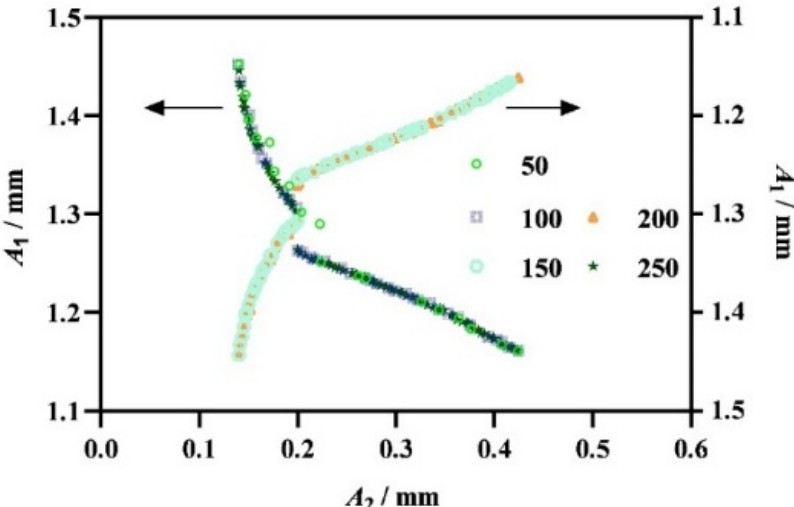

**Figure 9.** The effect of the initial population.

Figure 10 shows the effect of genetic generation on optimal accuracy, and the optimal results of (*A*1, *A*2) were both (1.45, 0.14,) and (1.16, 0.42) when the genetic generation

number was 400, 800, 1200, 1600, and 2000, separately. Therefore, the effect of genetic generation was not obvious. Meanwhile, the generation number of calculation termination was 143, 114, 113, 132, and 122, separately, according to the above setup. The generation number of calculation termination decreased at first and increased with the increase in generation numbers. In order to obtain a higher calculation efficiency, the generation number of 1200 was more suitable for the optimal model.

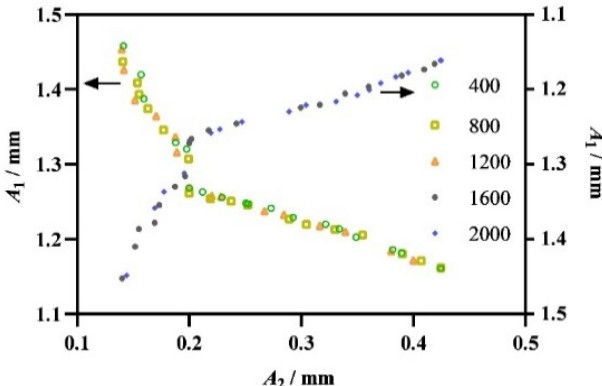

**Figure 10.** The effect of genetic generation.

Figure 11 shows the effect of mutation and the crossover rate on optimal accuracy and termination generation, in which the combination of mutation and crossover rate, (0.002, 0.1), (0.006, 0.3), (0.009, 0.45), (0.01, 0.5), (0.05, 0.7), and (0.09, 0.9) was separately selected to optimize the model and was numbered 1, 2, 3, 4, 5, and 6, respectively. According to the calculated results, the optimal result fluctuated with the change of mutation and crossover rate. The values of $A1$ and $A2$ were smaller than those of the other groups when the combination was (0.05, 0.7). Meanwhile, the two groups of higher computational efficiency were separately (0.006, 0.3) and (0.05, 0.7). Therefore, the combination of (0.05, 0.7) was more appropriate according to the above analysis.

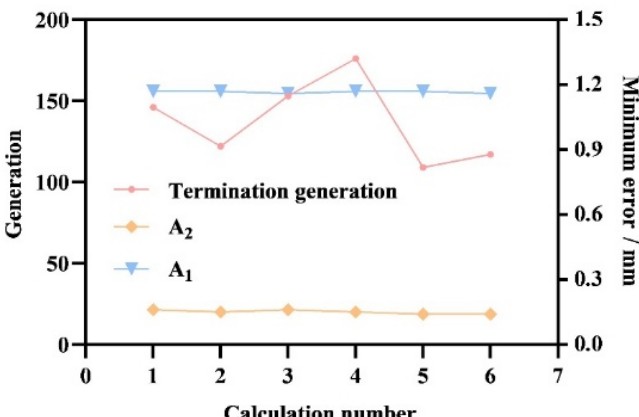

**Figure 11.** The effect of mutation and crossover rate.

In summary, the optimal values of ($A1$, $A2$) were opposite, and the upper fillet error could be compensated for by adjusting unsupported distances according to the feature of the forming process, and the lower fillet error was only improved through optimal process parameters. Therefore, $A2$ should be less than 0.5 mm, and a range of less than 0.3 mm was selected to eliminate the effect of $A1$ adjusted on $A2$. According to the above analysis, the combination of (1.23, 0.27) and ($A1$, $A2$) was viewed as an optimal result, and the corresponding forming process parameters ($a$, $b$, $c$) were separately 0.738, −0.098, and 0.04.

### 3.3. Effect of Unsupported Distances on A1

Figure 2 shows the stress distribution of the forming process. The unsupported region had a sinking defect due to the effect of the forming region. Therefore, the value of $A1$ was mainly determined by the unsupported distance. Figure 12 shows the forming profile of the unsupported region in the initial forming stage of EHIF, in which $A1$ was gradually increased with the increase in unsupported distance. Therefore, five distances, 0.5 mm, 1 mm, 1.5 mm, 2 mm, and 2.5 mm, were adopted to analyze the effect of unsupported distances on $A1$, and the aforementioned optimal forming process parameters were used for the five experiments.

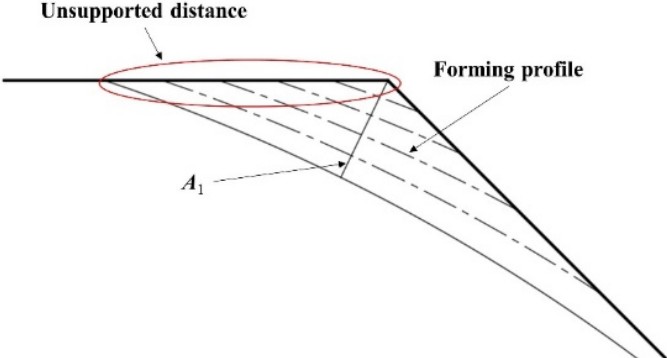

**Figure 12.** Sketch of the forming profile of unsupported regions.

Figure 13 shows the effect of unsupported distances on $A1$ and thickness, in which the value of $A1$ rapidly increased in the unsupported distance range of [0.5, 1) and (2.0, 2.5], and the change of $A1$ was gradual in the unsupported distance range of [1, 2.0]. Meanwhile, the value of $A1$ was less than 0.5 mm in the unsupported distance range of [0.5, 2.0]. However, the thickness of forming parts rapidly reduced when the unsupported distance was less than 1 mm, and the crack defect could be obtained for the forming part. In order to ensure allowable errors and large safety distances, the unsupported distance of 2 mm was adopted to fabricate the part designed. According to the above optimal and compensating parameters, the part with Ti-6Al-4V titanium alloy was fabricated and its profile was obtained through wire-electrode cutting. Figure 14 shows the error of forming part based on the optimal scheme, and the errors between the forming profile and the design profile were both less than 0.5 mm. Therefore, the forming scheme was reliable based on the optimal process parameters and reasonable unsupported distances.

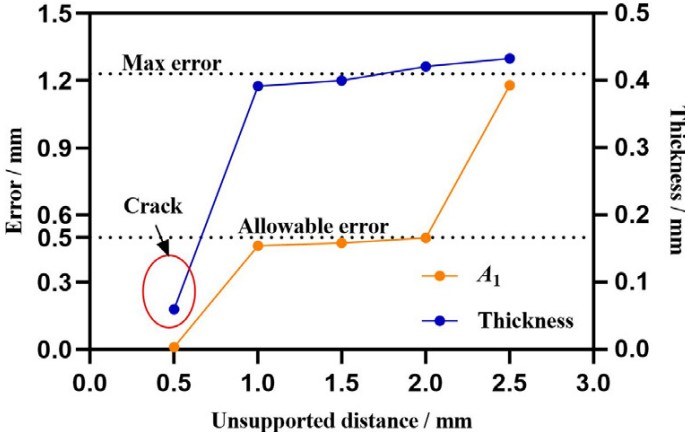

**Figure 13.** The effect of unsupported distances on $A2$ and thickness.

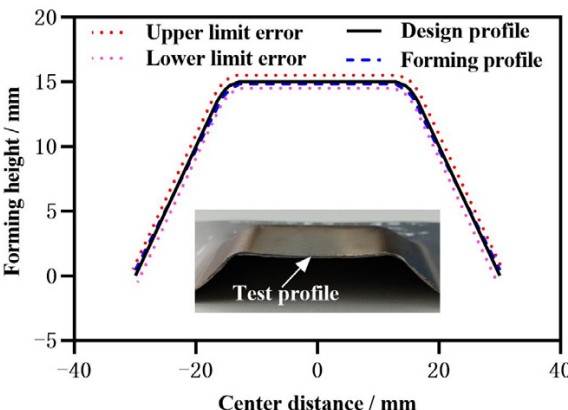

**Figure 14.** The comparative analysis between formed and designed profiles.

In order to verify the engineering applications of the optimization scheme, an engineering part with Ti-6Al-4V titanium alloy was fabricated based on the aforementioned optimal forming scheme ($a = 0.738$, $b = -0.098$, $c = 0.04$, and the unsupported distance of 2 mm), and its fabricating process is shown in Figure 15. Figure 16 further shows the profile error for the part, in which the transverse and longitudinal profile errors were both less than 0.5 mm. Therefore, the above optimal scheme was feasible and had a greater potential for engineering applications.

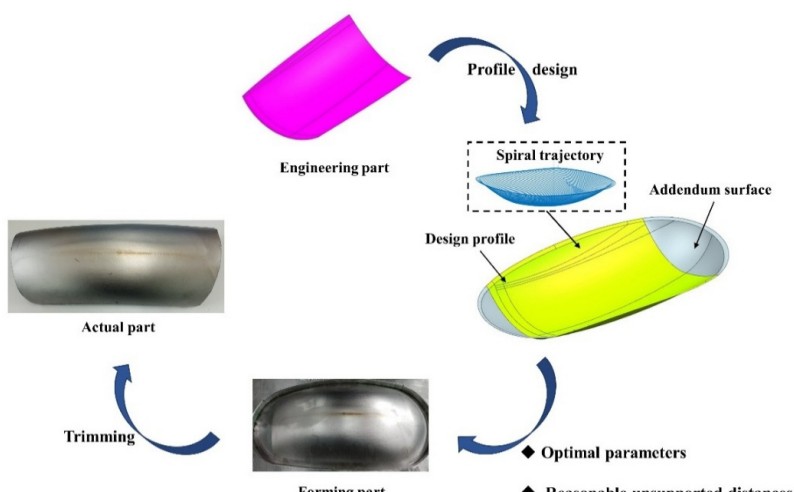

**Figure 15.** Sketch of the fabricating process of the engineering part.

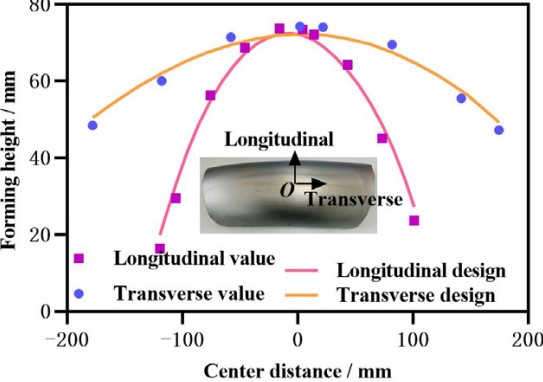

**Figure 16.** The error analysis of engineering parts.

## 4. Conclusions

In order to obtain a higher forming accuracy part, a combined optimization of process parameters and unsupported distances was proposed to control the dimensional accuracy of forming parts in EHIF. Two errors, $A1$ and $A2$, were viewed as optimal targets, and the corresponding predictive models were established based on the response surface methodology. Meanwhile, the relation between $A1$ and $A2$ was opposite, and the error of $A2$ was difficult to compensate for by adjusting unsupported distances. Therefore, the minimum value of $A2$ was firstly selected, and then the value of $A1$ could be obtained based on the corresponding $A2$. On this basis, the optimal process parameters (a, b, c) were separately 0.738, $-0.098$, and 0.04 were obtained through NSGA-II, and a combination of (1.23, 0.27) and ($A1$, $A2$) was obtained according to the relation between $A1$ and $A2$. Meanwhile, the effect of the unsupported distance on $A1$ was significant, and the unsupported distance of 2 mm was adopted to control the value of $A1$, so that the value of $A1$ was less than 0.5 mm. Finally, the square cone and the engineering part were separately fabricated based on the optimal scheme, and the profile errors of the parts were both less than 0.5 mm. In conclusion, the combined optimization of process parameters and unsupported distances produced a part with high dimensional accuracy, and the method proved significant in its potential engineering applications. In the future, the optimal method proposed could be further extended for the hot incremental sheet forming of light alloys, and the dimensional accuracy of complex parts could be studied based on the optimal method.

**Author Contributions:** Z.L.: conceptualization, investigation, formal analysis, and writing—original draft. S.H.: investigation and formal analysis. Z.A.: investigation and writing—review and editing. Z.G.: investigation and writing—review and editing. S.L.: formal analysis and funding acquisition. All authors have read and agreed to the published version of the manuscript.

**Funding:** This work was supported by The National Natural Science Foundation of China (Grant No. 22272013 and Grant No. 52205374), The Special Basic Cooperative Research Programs of Yunnan Provincial Undergraduate Universities' Association (Grant No. 202101BA070001-260), The Scientific and Technological Research Program of Chongqing Science and Technology Bureau (Grant No. cstc2021jcyj-msxmX1047), The Talent Introduction Project of Kunming University (Grant No. XJ20210033), and The Scientific Research Fund Project of Yunnan Provincial Department of Education (Grant No. 2022J0636).

**Institutional Review Board Statement:** Not applicable.

**Informed Consent Statement:** Not applicable.

**Data Availability Statement:** The data presented in this study are available on request from the corresponding author.

**Conflicts of Interest:** The authors declare no conflict of interest.

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
