# Peer review of "Multi-Objective Optimization of Dimensional Accuracy in Electric Hot Incremental Sheet Forming"

_coatings, doi:10.3390/coatings13050923_

Round 1

Reviewer 1 Report

Question 1: There are some grammatical and type errors in the paper; the paper should be free from all errors.

Response: The reviewer’s comment is very instructive. The grammatical and type errors of the article have been modified in the manuscript revised.

Question 2: Polish the abstract by presenting some significant and key outcomes.

Response: The reviewer’s comment is very instructive. The abstract of the revised manuscript has been modified.

Question 3: The introduction section should have some most recent and new publications. Improve the novelty of the work.

Response: The reviewer’s comment is very instructive. The introduction section of the revised manuscript has been modified.

Question 4: How are Eqs 1 and 2 formed? Give some details.

Response: The reviewer’s comment is very instructive. The formed procedure of Eqs. 1 and 2 is provided in the revised manuscript.

Question 5: Give some references for all the tables, if any.

Response: The reviewer’s comment is very instructive. The formed procedure, namely Ref. 27, of tables has been provided.

Question 6: Recheck Figures 8-13, and 15 and their relevant discussion.

Response: The reviewer’s comment is very instructive. Figures 8-13, and 15 have been rechecked, and the relevant discussion has been revised.

Question 7: Revise the results and discussion section briefly; I think you are missing some results here. Also, give some detail about the figures.

Response: The reviewer’s comment is very instructive. The results and discussion section has been modified in the revised manuscript.

Question 8: Give some justification for the methodology used for this research.

Response: The reviewer’s comment is very instructive. The methodology is proposed base on the forming process defect of Ref. 1 and 2.

Question 9: Include more outcomes of your study in the conclusions and future directions.

Response: The reviewer’s comment is very instructive. The conclusions section has been modified in the revised manuscript.

Reviewer 2 Report

Question 1: In the abstract, the authors often use the word "multipurpose". So they write: "the predictive model of multi-objective errors", "the multi-objective optimal model", "the multi-objective optimal method". At the same time, it is clear from the text of the article that there is neither an optimal model nor an optimal method in this work. There can be one goal, minimization or maximization of some generalized two-factor accuracy indicator. Here the authors can, in order to improve the quality of work, it is required to formalize such an indicator, where it is possible to clarify the importance of A1 and A2, using special weight coefficients.

Response: The reviewer’s comment is very instructive. A1 and A2 are separately two region errors of the forming part. Therefore, there are two targets for the part. Meanwhile, NSGA-II is used to obtain the minimum value of A1 and A2, and then the optimal model is a multi-objective model. The relevant expression has been modified in the revised manuscript.

Question 2: The introduction is written extremely poorly. It is not permissible to make generalized references to publications from 1 to 10 and from 11 to 16, while having only 21 sources in the list of sources used. There is no adequate analysis of the research conducted earlier in this area, where the authors point to the strengths and weaknesses of a particular work. In the introduction, the authors do not set an adequate goal for their own research, instead they give a description to Figure 1 - "a reasonable scheme of formation", which is not appropriate in the introduction, and also implies the unreasonableness of all other schemes before this one. Figure 1 and its explanation can be moved to the second section of the work.

Response: The reviewer’s comment is very instructive. The introduction section of the revised manuscript has been modified, and the corresponding structure of the article has also been revised.

Question 3: Figure 4 needs to be described in more detail in the text of the article, since it contains the main idea of the work.

Response: The reviewer’s comment is very instructive. The relevant expression of Figure 4 has been further described in detail.

Question 4: Conclusions on the work need to be rewritten. The lack of calculation of a multifactorial accuracy indicator does not allow us to talk about the optimality of the decisions made, we can talk about improving the accuracy of forming parts, but this accuracy should be shown by a generalized indicator.

Response: The reviewer’s comment is very instructive. The conclusions section has been modified in the revised manuscript.

Question 5: The list of sources is small, it needs to be increased by reflecting this in the expansion of the introduction.

Response: The reviewer’s comment is very instructive. The relevant suggestions have been modified in the revised manuscript.

Round 2

Reviewer 1 Report

I recommend the manuscript for Publication.

Reviewer 2 Report

The authors have done a good job, which in my opinion has significantly improved the quality of the article.